# Automated Classification of Cognitive Workload Levels Based on Psychophysiological and Behavioural Variables of Ex-Gaussian Distributional Features

**DOI:** 10.3390/brainsci12050542

**Published:** 2022-04-23

**Authors:** Monika Kaczorowska, Małgorzata Plechawska-Wójcik, Mikhail Tokovarov, Paweł Krukow

**Affiliations:** 1Department of Computer Science, Lublin University of Technology, 20-618 Lublin, Poland; m.kaczorowska@pollub.pl (M.K.); m.plechawska@pollub.pl (M.P.-W.); m.tokovarov@pollub.pl (M.T.); 2Department of Clinical Neuropsychiatry, Medical University of Lublin, 20-439 Lublin, Poland

**Keywords:** ex-Gaussian modelling, classical machine learning, cognitive workload

## Abstract

The study is focused on applying ex-Gaussian parameters of eye-tracking and cognitive measures in the classification process of cognitive workload level. A computerised version of the digit symbol substitution test has been developed in order to perform the case study. The dataset applied in the study is a collection of variables related to eye-tracking: saccades, fixations and blinks, as well as test-related variables including response time and correct response number. The application of ex-Gaussian modelling to all collected data was beneficial in the context of detection of dissimilarity in groups. An independent classification approach has been applied in the study. Several classical classification methods have been invoked in the process. The overall classification accuracy reached almost 96%. Furthermore, the interpretable machine learning model based on logistic regression was adapted in order to calculate the ranking of the most valuable features, which allowed us to examine their importance.

## 1. Introduction

Cognitive load classification is the subject of numerous scientific studies [1,2,3]. Its goal is to verify whether there are objective indicators of several cognitive workload levels enabling its recognition by various computational methods. The authors carried out both subject-dependent [4] and subject-independent [5] classification inquiries; the latter becoming progressively more popular in the research. The subject-independent classification approach is where samples taken from a single subject are fully included in one dataset (train or test) in order to ensure reliability of results. In the literature, one can also find utilization of both approaches applied simultaneously [6]. The most common classification of the level of cognitive load is the binary approach (presence or absence of overload) [7], but there are also attempts to classify several levels of cognitive load [8]. The literature review shows that the cognitive load is most often tested on the basis of eye-tracking signals [8] and EEG [9]. As for classification methods, both classic classifiers and deep neural networks [10,11] are administrated. In terms of classical classifiers one can mention such models as support vector machine (SVM) [1,12], linear discriminant (LDA) [3] and k-nearest neighbours (kNN) [13]. Depending on the type of classification, the authors usually present their results between 70–95% accuracy. Yamada and Kobayashi [14] conducted a study including 12 participants and undertook a two-class classification of cognitive load with the use of the SVM classifier, achieving a result of over 90% accuracy. The model created classified the effort or its lack, regardless of the age of the examined person. In other research [15], the authors presented a seven-class classification, which concerned the sorting of cognitive load on the basis of the eye-tracking signal when performing arithmetic tasks, from the easiest to the most difficult. The authors obtained an accuracy reaching from 0.4 to 0.98 using artificial neural networks. There are other studies presenting the results of binary [16,17,18] and three-class classification [6,19] of cognitive workload. As suggested earlier, the classification studies on cognitive workload have two main purposes: to find the objective indicators of cognitive overburden (e.g., behavioural, cognitive and physiological) and to verify computational methods enabling the most accurate automatic recognition and differentiation of overload levels.

The vast majority of cognitive experimental data sets consisting of a large amount of numeric results are still analysed with the application of parametric analytical methods, first of all in the form of means and standard deviations. Despite this, researchers focused on measuring reaction times (RTs) have noticed that the distribution of such outcomes, even when collected from healthy participants, is often skewed and contrary to expectations; the empirical distribution of RT data does not always fit the Gaussian model on which parametric statistics are based [20]. Typically, the skewness of the distribution of the RT series is positive; that is, it has a set of the most typical and at the same time relatively short, most frequent RTs, and an extended right-side convolution tail, containing the rarest, but at the same time longest RTs, representing the most unusual, prolonged RT outliers of the whole distribution [21]. Such a form of data distribution suggests that in a series of RTs parametric symmetrical standard deviation (SD) does not fully cover the real range of outliers, and instead, averaging all data to the form of arithmetic means eliminates some portion of uncommon observations and ultimately makes it impossible to assess the true extent of intra-individual variability (IIV). Hence, to circumvent some limitations of the parametric approach to experimental RT-based data, the so-called ex-Gaussian methodology has been developed, allowing analysis of positively skewed distribution so as not to eliminate outliers, and at the same time, not distorting the range of typical results. Ex-Gaussian distribution modelling provides quantitative characteristics in the form of three independent parameters: mu (*μ*), representing the mean of the normal component and reflecting average performance or the most frequent results; sigma (*σ*), defining the symmetrical standard deviation of the normal component; and tau (*τ*), covering the exponential part of the distribution with the most prolonged and most often rarest RTs [22]. In cases when the experimental procedure consists of displaying repetitive stimuli and/or relatively unified reactions are expected, it is considered that the *τ* parameter covering the scope of the most prolonged RTs might be understood as an indicator of “attentional lapses” or “off-task mind wandering”. Attentional lapses are usually due to transient failures in performance controlling mechanisms occurring, for example, due to increasing cognitive fatigue [23,24,25]. RT-based and other studies focused on cognitive performance and its disturbances confirm that an increased range of attentional lapses and off-task mind wandering is associated with lower levels of effortful control [26], states of reduced alertness and generally diminished productivity [27,28], and additionally with mental health risk factors, such as anxiety and negative affect [29,30].

Although measures of intra-individual variability, including ex-Gaussian modelling, have so far been used mainly to analyse features of RT distribution, it does not mean that these methods cannot be successfully implemented for other types of empirical data. For example, indicators of intra-subject variance were used to analyse neuroimaging results, especially regarding patterns of variability in neuronal functional connectivity matrices [31,32], analogous computation has been exploited in studies on heart rate variability [32,33,34] and ex-Gaussian modelling enabled the highlighting of intergroup differences in the extremely high measures of IgG allergic reaction markers in patients with depression, irritable bowel syndrome and healthy controls [35].

It can be argued without a doubt that, just as the application of ex-Gaussian modelling to cognitive data in the form of RT series is relatively common, the use of this quantitative analysis method for eye-tracking output is still scarce. To our knowledge, one of the first studies in which the distribution of variables such as fixation length and intersaccadic intervals were successfully matched to ex-Gaussian distribution was carried out by Otero-Millian et al. [36]. In the following years, the ex-Gaussian modelling of eye-tracking results was described only a few more times, while probably the fullest theoretical and empirical justification for the application of distributional analyses to parameters such as fixation length was presented by Guy, Lancry-Dayan and Pertzov [37]. These authors documented not only the parametric mean of fixation duration (FD), but also that its empirical distribution is sensitive to experimental manipulations of the task input; they also alluded to the results of previous studies which confirmed the existence of significant relationships between task features such as semantic clarity, stimuli familiarity and cognitive individual differences, e.g., regarding working memory, and FD ex-Gaussian characteristics, especially *μ* and *τ*. First of all, in their original study Guy and co-workers evidenced that under three different experimental conditions FD distributions fitted the ex-Gaussian curves even better than the Gaussian one; *μ* and *τ* cover significantly different aspects of eye movements and are not redundant or overlapping variables. Additionally, when the same individuals were subjected to the same eye-tracking experiments with a 7-day interval, ex-Gaussian parameters exhibited very high reliability (the correlation between the first and second assessments was at least 0.80). In the end, Guy et al. [17] argued that the *τ* parameter is associated with repetitive exposures to the same images, while *μ*, as previously suggested, is to a greater extent related to stimuli familiarity and individual differences in problem-solving efficiency. The increase in τ calculated from FD might represent a psychophysiological marker of attentional lapses because as regards the repetitive presentation of stimuli, i.e., an experimental situation, when a testee already knows given stimuli and the duration of individual fixations enlarges, it is rather unlikely that such extremely prolonged fixation represents the process of stimulus decoding since it has already been visually decoded. Although in the discussed research ex-Gaussian modelling was applied only to one variable, which, as RTs, has a temporal character (duration), we postulate that since the distribution has an axis representing the frequency of measured observations, this approach may also be used to analyse non-temporal eye movement features, such as the magnitude or amplitude of microsaccades and saccades. Then, *μ* might cover the typical, most recurring observations, while τ will refer to the rarest and to some extent extreme attributes of eye-movements. Taken together, we assume that application of ex-Gaussian modelling to data collected from eye-tracking during exposure to tasks eliciting cognitive workload seems to be fully justified. Additionally, we expect that both μ and τ measures will prove to be significant predictors allowing objective classification of cognitive workload levels with high accuracy.

At this point, we would like to assert that according to our knowledge, although the eye-tracking data were analysed with an application of ex-Gaussian modelling, it was not implemented regarding data taken from the cognitive workload experiment. We presume, that increasing cognitive overburden is associated with a growing number of atypical RTs and atypical physiological events, and therefore, utilization of ex-Gaussian parameters may enhance the possibilities of its automated classification.

In the current study, we implemented ex-Gaussian modelling to analyse two types of data: cognitive variables in the form of RTs and correct or invalid reactions together with psychophysiological variables collected from the eye-tracker. Eye-tracking values cover data related to saccades, fixations and blinks. Both of these groups of data were gathered during an experimental procedure whose aim was to elicit the state of cognitive overburden. In comparison with the aforementioned Guy et al. study [17], we decided to corroborate our premises according to which ex-Gaussian characteristics might reflect not only time-related variables’ distributions (e.g., fixation duration) but also another type of psychophysiological measures collected during the cognitive workload experiment. This might be acknowledged as potentially original input in our research.

Considering the above, our study has two main goals:To verify whether the cognitive and physiological data collected during cognitive-workload-related experiments fit the ex-Gaussian distribution;To determine the possibilities of machine-learning-based classifiers regarding automatic recognition of cognitive workload using ex-Gaussian parameters of eye-tracking and cognitive measures.

Taking into account the above goals, this manuscript is organized as follows: The Materials and Methods section describes the research procedures covering the computer application, equipment, experiment details, data processing, classification methods and statistical procedures. The Results section contains distributional analyses regarding the fitness of the obtained data to ex-Gaussian curves and classifications outcomes. The last section (Discussion) describes and explains the results with reference to the study goals.

## 2. Materials and Methods

### 2.1. Research Procedure

Obtained experimental input collected in the current study is fully original and does not coincide with our previous research. The experiment itself consisted of administrating a computerised version of the digit symbol substitution test (DSST) [38], which measures the speed of cognitive processing. While performing DSST, a subject’s task was to connect the abstract symbols to the corresponding numbers as quickly as possible. In order to match the number with the symbol, the participant had to indicate with the mouse an appropriate number on a displayed keyboard. The experiment was divided into three parts. In each part, the participant’s task was to match the number with the symbol, but the parts differed regarding the number of symbols and duration of the task itself. The first part lasted 90 s and had 4 symbols to choose from, the second and third parts had 9 symbols, but the second part lasted 90 s and the third part lasted 180 s. Thus, it can be assumed that the level of cognitive load in each successive part was higher than in the previous one, giving in sum three levels of cognitive workload. The application was written in Java 8.0.

Figure 1 shows the procedure of the experiment. Each part consisted of the same steps. The successive parts differed in the difficulty of the task being performed.

### 2.2. Data Acquisition

The experiment was carried out in the laboratory using the proprietary application and the stationary eye-tracker Tobii Pro TX300 (Tobii AB, Stockholm, Sweden). The eye activity-related data were recorded with 300 Hz frequency. The Tobii Studio 3.2, a software compatible with the eye-tracker, was used in the described experiment. In general, the experiment lasted about 15 min. A nine-point calibration was performed at the start of each measurement. Then the participant proceeded to implement each of the parts. Before each part, the trial version was presented so that the participant knew what to do in a given part. Using the Tobii Studio 3.2 software, the eye activities were exported and saved to an individual file. Additionally, the application generated a file with information about the answers given by the participant. The accepted quality of eye-tracking recording was set to 90% of eye activity.

Thirty healthy participants (students) were involved in the study. The group included subjects aged 20 to 24 (M = 20.45; SD = 1.62); 23 males and 7 females. Additionally, participants had no history of psychiatric nor neurologic diagnoses and reported not to be undergoing medical treatment or taking medication. All participants had normal/corrected-to-normal vision.

### 2.3. Data Processing

The dataset applied in the study covered features related to eye-activity and cognitive results of the DSST performance. The following eye-activity-related measures, obtained from the eye-tracking equipment, were used in the study:Saccades [39], understood as eyes movements bringing the essential visual information onto the most sensitive part of the retina. This process is performed to retrieve information easily [40].Fixations [39], described as the time period when the visual information is proceeded. During that time eyes stay in a relatively stable position.Blinks, identified by Tobii Studio software as zero data saccades [41].

Among the cognitive measures received from the DSST application we applied the following:Response time defined as the time needed to perform a single matching in the application.Good response numbers understood as the number of correct answers given in a certain time period.

The abovementioned variables were used in the data processing procedure, which covered such steps as data synchronisation, feature extraction, feature selection and classification.

The synchronisation procedure was related to the process of merging of data received from eye-tracker equipment and from the DSST application. Synchronisation was made on the basis of timestamps saved in both datasets. As each participant took part in three parts of the assessment, each with a different level of cognitive workload, 90 observations were included in the final dataset.

Synchronised outputs were subjected to the basic data cleansing procedure. One highest and one lowest value were deleted from each data series obtained from a particular subject.

The feature extraction procedure was performed using ex-Gaussian statistics. We decided to use this method as it offers a good prognosis [35] for dissimilarity detection in groups. Ex-Gaussian parameters allow us to consider the exponential specificity of the data. The ex-Gaussian distribution enables us to distinguish three independent parameters:Mu (*µ*)—corresponding to the mean of the normal component;Sigma (*σ*)—representing the symmetric standard deviation of the normal component;Tau (*τ*)—reflecting the exponential part of the distribution.

Mu and sigma in ex-Gaussian modelling correspond to classical mean and standard deviation. Mu results were used in the final dataset in order to consider averaged, most frequently occurring results. The tau parameter was included as it indicates the extremes in results dispersion, or outliers usually eliminated in evaluations based on normal distributions. Ex-Gaussian modelling was accomplished using the MATLAB toolbox “DISTRIB” and the recommendations of Lacouture and Cousineau [22].

Ex-Gaussian parameters were calculated for the following eye-related measures: saccades (amplitude of saccade, number of saccades and saccade duration), fixations (number of fixations and fixation duration) and blinks (number of blinks) as well as for DSST-related measures: number of correct answers and single trial response time. Among the listed features, the number of saccades, fixations, blinks and correct responses were extracted for the specific time intervals (10 s period). The taking into account of cognitive and psychophysiological data extracted for shorter intervals was dictated by the conclusions of our earlier research showing that information processing is specifically time-organised and shows variable dynamics and temporal changes in its efficiency [42,43,44]. Therefore, we decided that isolating the potential dynamic dimension of task performance may also in this case strengthen the effectiveness of classification of cognitive workload at various levels.

In the feature selection procedure the nonparametric Friedman test (α = 0.05) was applied to check the significance of the features. Results showed that most of the sigma parameters (for such features as number of fixations, fixation duration, number of saccades, saccade duration, number of correct answers and single trial response time) were non-significant. Taking into account these results and the fact that the sigma parameters indicate only how far the data are spread from the mean, we decided to discard all the sigma parameters from the further analysis in order to reduce the data dimensionality. Ultimately the resulting dataset contained the following 16 features:Saccade-related features: mu of saccade amplitude, tau of saccade amplitude, mu of saccade duration, tau of saccade duration, mu of saccade number in 10 s, tau of saccade number in 10 s;Fixation-related features: mu of fixation duration, tau of fixation duration, mu of fixation number in 10 s, tau of fixation number in 10 s;Blink-related features: mu of blink number in 10 s, tau of blink number in 10 s;DSST-related measures: mu of correct answers number in 10 s, tau of correct answers number in 10 s, mu of single trial response time, tau of single trial response time.

Additionally, correlation between particular features were checked using Spearman’s correlation coefficient with a significance level of α = 0.05. There were no significant strong correlations found in the dataset.

The classification procedure was performed in order to distinguish between three classes: low, medium and high cognitive workload. These three classes correspond to three parts of the DSST-based experiment. The dataset was randomly divided into the training and testing parts in the ratio 80:20. A subject-independent approach was applied in the procedure and data from a single participant were assigned to only one dataset (train or test) in order to ensure full independence of both datasets. Several classification algorithms such as decision tree, SVM, random forest and logistic regression classifier were applied in the study. Furthermore, feature weights were extracted by using the logistic regression model in order to assess the importance of particular features in the process of distinguishing cognitive workload levels. The entire train–test division and classification procedure was independently repeated 200 times in order to achieve reliable classification results.

## 3. Results

### 3.1. Distributional Analyses

The first step of the outcome analysis consisted of checking whether the data fit the ex-Gaussian distribution. As depicted in Figure 2, the majority of the studied variables’ empirical distribution matched to ex-Gaussian characteristics. In detail, Figure 2 presents distributions of measures such as amplitude of saccade, number of saccades, saccade duration, number of fixations, fixation duration, number of blinks, number of correct answers and single trial response time. Additionally, charts show the right-side exponential tail related to tau parameters.

The level of skewness was positive and constantly grew for measures at all cognitive workload levels, e.g., saccade duration (values 0.2231, 0.4379 and 0.4522, respectively) and response time (values 3.7235, 3.8627 and 4.0802, respectively).

### 3.2. Classification Results

A tree-class cognitive load classification was carried out. The quality of classification was measured with F1 and accuracy, which provided similar results. The following classifiers have been tested: decision tree with the Gini splitting criterion and unlimited depth, kNN with k = 5, SVM with a quadratic kernel, SVM with a cubic kernel, SVM with a linear kernel (C = 1, gamma = 1/16, other values of hyperparameters were tested in the course of preliminary experiments; however, no notable influence of classification performance was registered), logistic regression, random forest with 100 trees with the Gini splitting criterion as well as multilayer perceptron with two hidden layers and the ReLU activation function. The best obtained results are presented in Table 1. The first column contains the name of the classifier and the second contains the average accuracy score and the standard deviation in parentheses. The third column includes the mean values of the F1 measure and the standard deviation in parentheses. The fourth column shows the number of features used for the classification selected on the basis of the feature ranking. The column contains the number of features selected from the ranking. The number of features was the same in case of accuracy and the F1 measure. As shown, the best results were achieved for the random forest classifier—almost 96% with 16 features. Four classifiers obtained results above 90%. The SVM classifier with linear kernel needed first 10 features which led to the set of 14 features contained in the union of all separate classifier feature sets. The approach allowed us to obtain a score above 91%.

Figure 3, Figure 4 and Figure 5 show how the accuracy of the classifier changes for the classifiers with respect to the number of features taken for analysis. Feature importance was obtained with logistic regression. The following models were tested: random forest, SVM with a linear kernel and decision tree. The standard deviation was marked as horizontal black lines. An interesting situation can be observed for the SVM classifier with a linear kernel. On the basis of just one feature, an accuracy of more than 82% can be obtained. As for the random forest classifier, the deviation value decreases with the increase of the feature numbers.

### 3.3. Feature Ranking

In addition to a quantitative influence on classification accuracy, interpretable machine learning methods allow us to obtain weights corresponding to the importance of particular features. The differences between feature importance weights can be quite low, hence a ranking containing sorted features can be an inadequate approach. A better approach would be to apply cluster analysis to divide features into three groups based on their importance: high, medium and low. The method of k-means was utilized for obtaining the three clusters. The presented values of weights were produced in the course of multiple repeated experiments including training a classifier model on a random sample drawn from the analysed dataset. The experiment was repeated 1000 times for each classifier model.

Figure 6 presents barplots of sorted feature importance weights divided into the mentioned groups. Each subplot corresponds to a separate level of cognitive workload. The k-means clustering was applied independently to separate workload levels. Two models were applied for calculating the features’ weights: linear SVM and logistic regression. The models have been chosen as the main feature selectors due to the fact that they are among the most popular interpretable machine learning algorithms [45,46]. A ranking has been created for each of the levels: low, medium and high. The higher the feature is in the ranking, the more important it is. It can be noticed that some features from the low and medium levels are common, just like those from the medium and high levels.

Table 2 presents the features belonging to the clusters of high and medium importance for all the levels of cognitive workload with the application of the linear SVM model. As Figure 6 shows, the subsets belonging to the specific clusters obtained by different models are very similar; the only differences are related to the following features: tau of fixation duration and mu of correct answers number in 10 s, which were included into the clusters of lower importance. Table 2 presents the features included in the clusters of high and medium importance by the SVM model with linear kernel. The detailed list of features belonging to the medium and high importance clusters are presented separately for linear SVM and logistic regression in Appendix A.

When analysing the importance of the features in Table 2, it can be noticed that the common feature for all workload levels is tau of correct number of answers in 10 s. As for the feature sets selected for low and medium cognitive workload, they overlap to a high degree. The most important eye-tracking-related features are mu of blink number in 10 s, mu of saccade amplitude, mu of saccade number in 10 s and tau of saccade amplitude, whereas the set of the most important cognitive measures includes mu of single trial response time, tau of correct answers number in 10 s and tau of single trial response times. 

Furthermore, a high cognitive workload has the set of distinct features that overlaps with the other levels to a notably lesser extent, which can be related to different cognitive processes corresponding to that level.

## 4. Discussion

The major aims of the study were to find out whether the cognitive and physiological data collected during cognitive-workload-related experiments fit the ex-Gaussian distribution, and to verify whether it is possible to efficiently perform three-class classification of cognitive workload levels using interpretable machine learning models. An independent approach was applied in the study, so that data from particular subjects were not separated into trial and the main test datasets. Such an approach ensures a better reflection of reality as in practice the trained model is usually applied to data that are taken entirely (without data division) from a new participant. An additional purpose was to obtain the feature interpretability, which is especially important in subject-independent classification.

The dataset applied in the classification was composed of two sets of features: eye-tracking and cognitive-based. Collected eye-tracking data related to such measurements as saccades, fixations and blinks; in other words, typical outputs gathered during an eye-tracking procedures. Additionally, cognitive features were gathered as a consequence of the DSST performance. Participants did not have any additional equipment; no devices had been directly attached to their bodies and did not restrict their movements or hinder their testing.

The feature extraction procedure was performed in such a way that features such as number of saccades, fixations or blinks and number of correct responses were extracted for 10 s intervals. The feature extraction procedure was focused on ex-Gaussian statistics, especially the mu and tau parameters. The sigma parameters were included in the initial analyses because most sigma parameters calculated for particular measures occurred to be insignificant; therefore, we discarded all the sigma parameters from further analysis.

An interpretable machine learning model was adapted in order to calculate the ranking of the most valuable features. This ranking was used to improve the classification results. Furthermore, it gives valuable information about the process of mental workload. Two models were applied to assign and to verify specific weights to separate features. These models are logistic regression with elastic net regularization and SVM with linear kernel. Two models were chosen in order to verify the stability of feature importance ranking. It allows the placing of feature importance quantitatively in the model.

The prominence of features can be examined in two ways. The first is an approach including the whole set of features, cognitive and eye-tracking measures together. The feature ranking was analysed separately for each cognitive workload level (low, medium and high), as presented in Figure 6. K-means cluster analysis was applied to divide features into three groups of different importance: high, medium and low. Further analysis of feature importance was conducted based on features from the first two clusters, rejecting the third due its features having the lowest importance. Specific weights of features are presented in Figure 6. In our opinion considering features of the third cluster might be confusing. The detailed list of features belonging to particular clusters separately for linear SVM and logistic regression are presented in Appendix A. The most important feature for all three levels turned out to be the tau of the correct number of answers (in 10 s). The most noticeable differences between the low and medium levels were the mu of single trial response time, the mu of saccade amplitude, the tau of saccade amplitude and the mu of saccade number (in 10s). The difference between the medium and high levels is particularly strong for the feature tau of the correct number of answers (in 10 s). Moreover, the tau of saccade amplitude is also placed high in the ranking. The second approach assumed the division into cognitive and eye-tracking features. The detailed ranking for these two sets is presented separately in Appendix A). The cognitive feature set contained four features; the most important of them was the tau of the correct number of answers (in 10 s). In the case of the eye-tracking feature set including 12 features, the mu of saccade amplitude and the tau of saccade amplitude were high in the ranking. Overall results of feature ranking indicate features related to saccades, the number of correct answers and single trial response time as the most valuable in the case study. Results show that features related to fixations and blinks were not as important as other eye-tracking features, especially saccades.

The results of the classification show that the features based on ex-Gaussian statistics allow us to carry out a multi-class classification. Eight classifiers were initially tested and four of them enabled us to achieve an accuracy higher than 90%. They are logistic regression, random forest, SVM with a linear kernel and decision tree. Obtaining more than 90% accuracy is possible with 13 features (logistic regression). However, obtaining a result equal to almost 96% is possible with the use of 16 features. This result was obtained for the random forest classifier.

In sum, we find that introducing ex-Gaussian distributional characteristics to cognitive and physiological data associated with different levels of cognitive workload classification was beneficial. This claim seems to be particularly justified considering that among eye-tracking-related variables with the highest classification powers the majority covers the tau parameter, for example, considering saccade number, saccade duration and its amplitude. This means that a distributional feature which is usually deleted when analysed according to a parametric approach turns out to distinguish to the greatest extent three levels of cognitive processing overburden. However, the fact that mentioned indicators of eye-movement were characterised by the tau metric, and not by mu, suggests that cognitive workload, as it is studied with eye-tracker methodology, is probably highly associated with the amount of non-typical rare dimensions of eye-movement, not by the most common ones. Therefore, we suggest that the accumulation of cognitive workload might be physiologically expressed by an increase in the scope of outliers, not only by changes related to the most common features of performance.

There are some limitations of our study, which should be addressed. The experimental group was not well balanced regarding sex, with a clear predominance of men. However, according to our knowledge, there is no strong and unequivocal support for the notion that there are substantial sex differences regarding eye-movements schemata observed during non-sexual visual content scanning [47]. Nevertheless, we plan on providing a better gender-balanced group in future studies. Additionally, the feature ranking and classification rate might be changed in the case of a differently constructed experiment. It is worth examining the influence of the type and the order of tasks on the overall classification results. Changes in the requirements of the task which was administered to elicit a gradual increase in cognitive workload consisted in both elongation of the performance time (difference between parts 1, 2 and 3) and an increase in the number of stimuli to be processed (difference between part 1, 2 and 3). Therefore, our study did not distinguish whether the increase in cognitive load was generated by performance length or the scope of task internal complexity indicated by the number of given stimuli. However, our goal was not directly related to the problem of the relationship between the specificity of the task and the level of cognitive load. We assumed that the increasing modification of both the complexity of the task and its length should provoke an increase in cognitive overburden. The data achieved indicate the accuracy of our premise.

## Figures and Tables

**Figure 1 brainsci-12-00542-f001:**
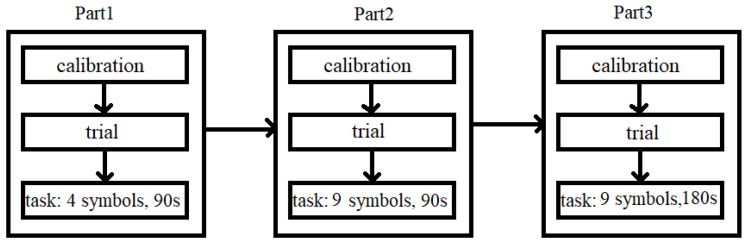
The procedure of the experiment.

**Figure 2 brainsci-12-00542-f002:**
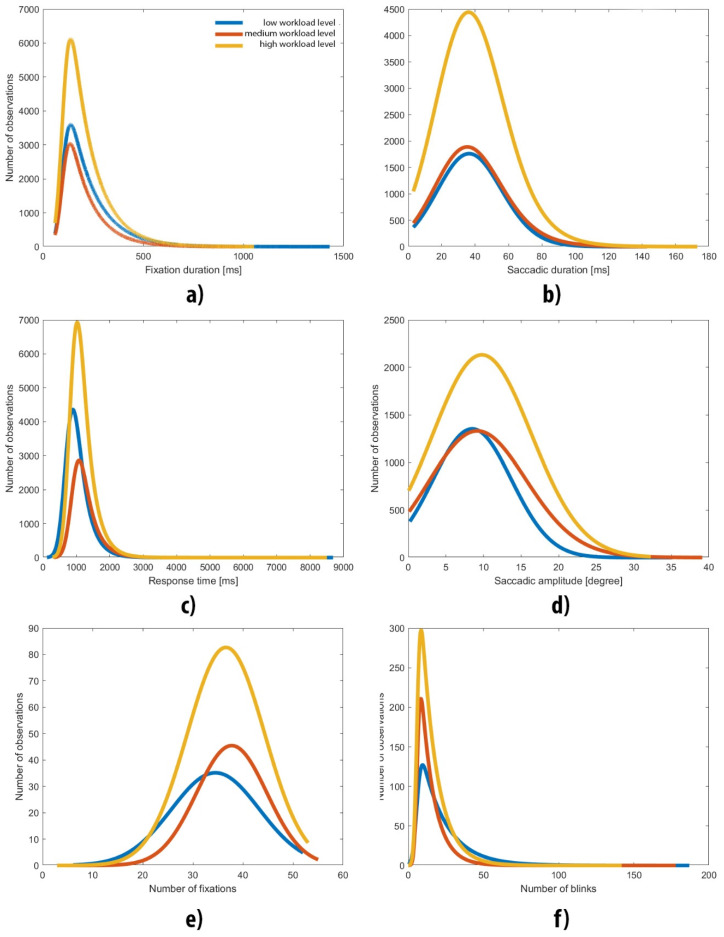
Distribution of features. Level 1 stands for low cognitive workload (**a**)—fixation duration, (**b**)—saccade duration, (**c**)—response time, (**d**)—saccade amplitude, (**e**)—number of fixations, (**f**)—number of blinks).

**Figure 3 brainsci-12-00542-f003:**
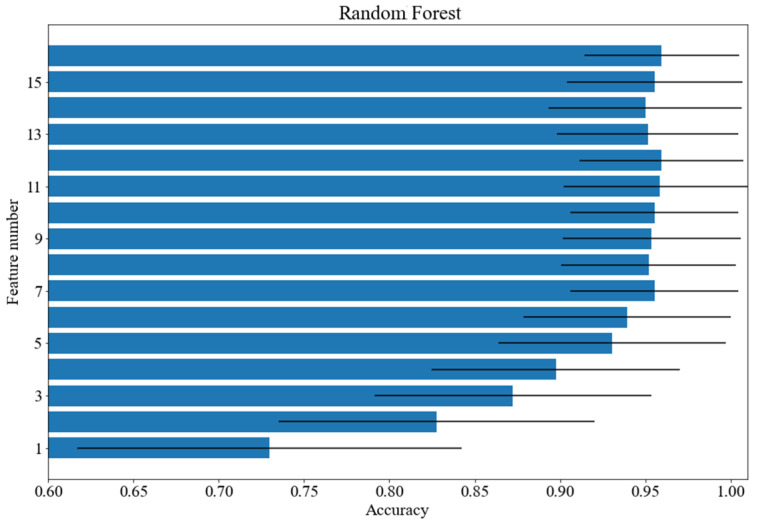
Accuracy scores for various number of features: random forest.

**Figure 4 brainsci-12-00542-f004:**
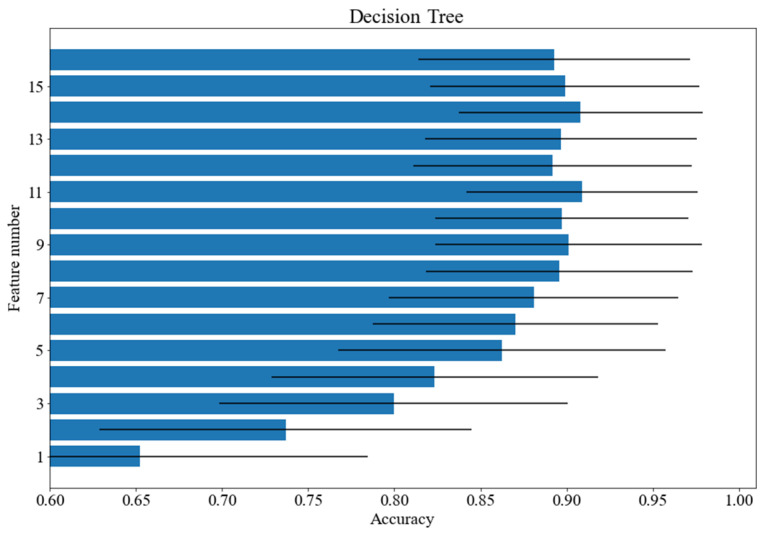
Accuracy scores for various number of features: decision tree.

**Figure 5 brainsci-12-00542-f005:**
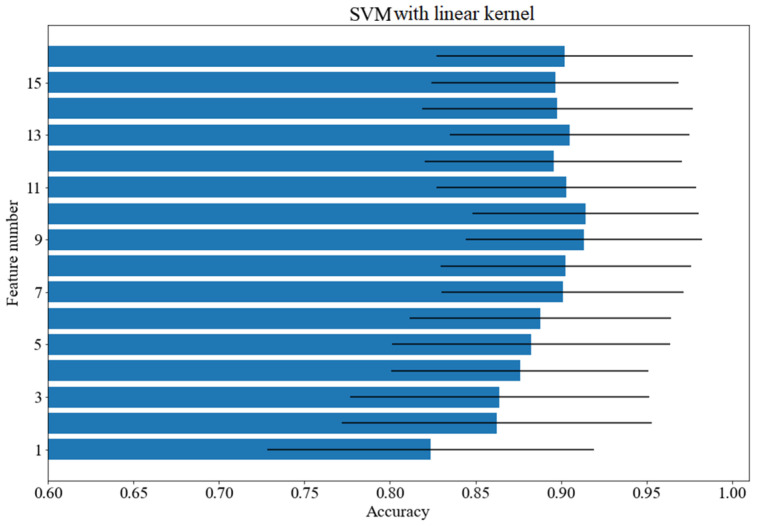
Accuracy scores for various number of features: SVM with linear kernel.

**Figure 6 brainsci-12-00542-f006:**
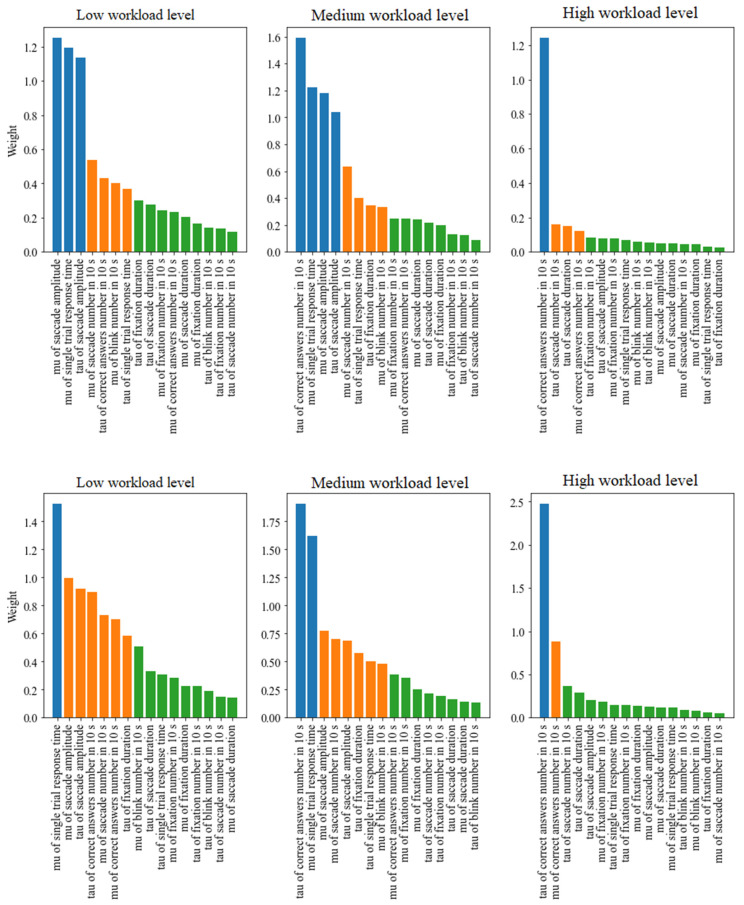
Clustered feature importance sorted in descending order. Top: SVM with linear kernel, bottom: logistic regression. Blue: high importance, orange: medium importance, green: low importance.

**Table 1 brainsci-12-00542-t001:** Classification results for a selected feature subset.

Classifier	Accuracy	F1	Number of Features
Decision Tree	90.91 (6.73)	90.93 (6.72)	14
SVM With Linear Kernel	91.44 (6.62)	91.36 (6.72)	14
Logistic Regression	90.06 (7.58)	90.03 (7.68)	13
Random Forest	95.97 (4.55)	95.98 (4.52)	16

**Table 2 brainsci-12-00542-t002:** The features belonging to the clusters of high and medium importance according to linear SVM model, presented separately for each cognitive workload level.

Low Cognitive Workload	Medium Cognitive Workload	High Cognitive Workload
mu of blink number in 10 s mu of saccade amplitude mu of saccade number in 10 s mu of single trial response time tau of correct answers number in 10 s tau of saccade amplitude tau of single trial response time	mu of blink number in 10 s mu of saccade amplitude mu of saccade number in 10 s mu of single trial response time tau of correct answers number in 10 s tau of fixation duration tau of saccade amplitude tau of single trial response times	mu of correct answers number in 10 s tau of correct answers number in 10 s tau of saccade duration tau of saccade number in 10 s

## Data Availability

The data presented in this study are available on request from the corresponding author.

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
