# Peer review of "Automated Classification of Cognitive Workload Levels Based on Psychophysiological and Behavioural Variables of Ex-Gaussian Distributional Features"

_brainsci, 2022, doi:10.3390/brainsci12050542_

Round 1
Reviewer 1 Report
- As opposed to the earlier version I reviewed, this paper has been thoroughly revised, and its flow and language have been significantly improved. However, there is still much room for improvement. See my comments below.
- In the manuscript, there are some minor rendering issues with mathematical symbols. Several times the symbol Ø appears without any context in the pdf version.
- Correct the spelling of the trial in Figure 1 instead of "trail".
- The legend names in Figure 2 should be added. Just labeling them as levels does not give much information.
- Describe the Hyperparameter values and how they were obtained for the SVM classification. Choosing suboptimal hyperparameters can explain the underperformance of the SVM with linear kernel.
- Logistic regression was selected for feature selection with weak justification. Table 1 shows that Logistic regression has the worst performance as a classifier. The paper should provide objective criteria for selecting an optimal model for feature ranking.
- As far as the validity of the feature ranking results is concerned, no quantitative evidence has been provided. I suggest testing the feature ranking with a randomization test, i.e., shuffling the data to create a null distribution.
- The objective of the paper is argued to "find out whether the cognitive and physiological data collected during cognitive-workload-related experiments fit the ex-Gaussian distribution, and to verify whether it is possible to efficiently perform three class classification of cognitive workload level using interpretable machine learning models." Despite this, a meaningful interpretation of the results remains elusive. A meaningful interpretation can be gained, for instance, by assessing the weights of the regression model and the beta values of the support vectors.
Author Response
- As opposed to the earlier version I reviewed, this paper has been thoroughly revised, and its flow and language have been significantly improved. However, there is still much room for improvement. See my comments below.
Thank you for your essential comments
- In the manuscript, there are some minor rendering issues with mathematical symbols. Several times the symbol Ø appears without any context in the pdf version.
The manuscript has been checked for linguistic and stylistic correctness. The symbol Ø appeared in order to indicate changes from the previous section (removing the word/phrase). In the new version, these symbols have been removed.
- Correct the spelling of the trial in Figure 1 instead of "trail".
It has been corrected, thank you.
- The legend names in Figure 2 should be added. Just labelling them as levels does not give much information.
It has been corrected, thank you.
- Describe the Hyperparameter values and how they were obtained for the SVM classification. Choosing suboptimal hyperparameters can explain the underperformance of the SVM with a linear kernel.
It has been added (please check p. 10)
- Logistic regression was selected for feature selection with weak justification. Table 1 shows that Logistic regression has the worst performance as a classifier. The paper should provide objective criteria for selecting an optimal model for feature ranking.
We have extended the section presenting the feature selection and the feature ranking procedures. We have added detailed rankings obtained for two different methods (linear SVM and logistic regression) (please see Figure 6). We have also performed cluster analysis in order to obtain features divided into three groups based on their importance: high, medium and low. As obtained results are comparable, we presented the features belonging to the clusters of high and medium importance for all the levels of cognitive workload with an application of the linear SVM model (please see Table 2). Minor differences to the results obtained for Logistic regression have been described in the text, whereas detailed list of features for Logistic regression is presented in Supplementary materials.
- As far as the validity of the feature ranking results is concerned, no quantitative evidence has been provided. I suggest testing the feature ranking with a randomization test, i.e., shuffling the data to create a null distribution.
Our procedure covered multiple repetitions (1000 repetitions for each classifier model), it was highlighted in the test (as previously we missed that information).
- The objective of the paper is argued to "find out whether the cognitive and physiological data collected during cognitive-workload-related experiments fit the ex-Gaussian distribution, and to verify whether it is possible to efficiently perform three-class classification of cognitive workload level using interpretable machine learning models." Despite this, a meaningful interpretation of the results remains elusive. A meaningful interpretation can be gained, for instance, by assessing the weights of the regression model and the beta values of the support vectors.
As it was described above, the analysis was extended and detailed numerical results have been provided. Figure 2 presents feature weights obtained for two weights. Additionally, detailed rankings with numerical weights also have been presented in Supplementary materials.
Reviewer 2 Report
Final spell check is required.
Author Response
Thank you for your comments. The manuscript has been checked for linguistic and stylistic correctness.
Reviewer 3 Report
The authors propose to describe eye movement and cognitive data collected during varying workload experimental trials using the ex-gaussian approach and to use the resulting parameters for automatic workload classification.
Two goals are stated:
- Verifying whether the ex-gaussian approach can fit the considered workload related data
- verify the effectiveness of ex-gaussian distribution parameters for workload data classification.
The first aim seems to be verified qualitatively, while a measure of the appropriateness of the fit is needed.
Please provide more details on the obtained ex-gaussian parameters describing the data used for classification. Present them in a table or in a figure.
Specific comments
Abstract. Line 4. "three levels of cognitive overburden" should be "... of cognitive workload"
Figure 1 reports a central block labeled "trail". Should it be "trial"?
Page 5. I don't understand the sentence "The ex-Gaussian statistics were calculated additionally converting the results in order to achieve the Gaussian distribution"
please rewrite the following: "From the each individual data series taken from each subject"
The legend for Figure 2 should be expanded. It would be more significant to show the data related to the features that will be selected for classification.
Page 9, second to last paragraph, second to last line. The sentence is not clear "On the other hand, among the features describing eye-tracking measures from the top ten for all features were the tau of saccade amplitude and the mu of blink number in 10 s"
P. 9, last paragraph, last line "feature number" should probably be "number of features"
Author Response
- The authors propose to describe eye movement and cognitive data collected during varying workload experimental trials using the ex-Gaussian approach and to use the resulting parameters for automatic workload classification.
Two goals are stated:
Verifying whether the ex-Gaussian approach can fit the considered workload related data
verify the effectiveness of ex-Gaussian distribution parameters for workload data classification.
The first aim seems to be verified qualitatively, while a measure of the appropriateness of the fit is needed.
Thank you for your comment. We have extended the section presenting the feature selection and the feature ranking procedures to present objective measures. We have added detailed rankings obtained for two different methods (linear SVM and logistic regression) (please see Figure 6). We have also performed cluster analysis in order to obtain features divided into three groups based on their importance: high, medium and low. As obtained results are comparable, we presented the features belonging to the clusters of high and medium importance for all the levels of cognitive workload with an application of the linear SVM model (please see Table 2). Minor differences to the results obtained for Logistic regression have been described in the text, whereas a detailed list of features for Logistic regression is presented in Supplementary materials.
- Please provide more details on the obtained ex-Gaussian parameters describing the data used for classification. Present them in a table or in a figure.
We have added all ex-Gaussian parameters used for classification. As this dataset is quite large, we have added it to the Supplementary materials (please see Table 7S).
- Specific comments
- Line 4. "three levels of cognitive overburden" should be "... of cognitive workload"
It has been corrected, thank you.
- Figure 1 reports a central block labeled "trail". Should it be "trial"?
It has been corrected.
- Page 5. I don't understand the sentence "The ex-Gaussian statistics were calculated additionally converting the results in order to achieve the Gaussian distribution"
We have decided to remove that sentence as it does not give much sense.
- please rewrite the following: "From the each individual data series taken from each subject"
It has been corrected, please check page 6.
- The legend for Figure 2 should be expanded. It would be more significant to show the data related to the features that will be selected for classification.
The legend has been changed, thank you.
- Page 9, second to the last paragraph, second to last line. The sentence is not clear "On the other hand, among the features describing eye-tracking measures from the top ten for all features were the tau of saccade amplitude and the mu of blink number in 10 s"
This paragraph has been rewritten and this sentence was removed.
- 9, last paragraph, last line "feature number" should probably be "number of features"
We have changed all occurrences in the manuscript.
Round 2
Reviewer 1 Report
With the suggested changes, the revised version of the paper tried to improve.